# LSP : Acceleration and Regularization of Graph Neural Networks via Locality Sensitive Pruning of Graphs

## Abstract

Graph Neural Networks (GNNs) have emerged as highly successful tools for graph-related tasks. However, real-world problems involve very large graphs, and the compute resources needed to fit GNNs to those problems grow rapidly. Moreover, the noisy nature and size of real-world graphs cause GNNs to over-fit if not regularized properly. Surprisingly, recent works show that large graphs often involve many redundant components that can be removed without compromising the performance too much. This includes node or edge removals during inference through GNNs layers or as a pre-processing step that sparsifies the input graph. This intriguing phenomenon enables the development of state-of-the-art GNNs that are both efficient and accurate. In this paper, we take a further step towards demystifying this phenomenon and propose a systematic method called Locality-Sensitive Pruning (LSP) for graph pruning based on Locality-Sensitive Hashing. We aim to sparsify a graph so that similar local environments of the original graph result in similar environments in the resulting sparsified graph, which is an essential feature for graph-related tasks. To justify the application of pruning based on local graph properties, we exemplify the advantage of applying pruning based on locality properties over other pruning strategies in various scenarios. Extensive experiments on synthetic and real-world datasets demonstrate the superiority of LSP, which removes a significant amount of edges from large graphs without compromising the performance, accompanied by a considerable acceleration.

## 1 Introduction

Graph neural networks have become extremely popular for tasks involving graph-data. The majority of architectures employ varieties of message-passings, such as Graph Convolutional Networks (Kipf & Welling, 2016), Graph Isomorphism Networks (Xu et al., 2018) and more (Wu et al., 2020). These architectures propagate belief information from nodes of a graph through adjacencies in order to generate representations that depend on wide graph environments. This property makes them well suited for learning tasks such as node classification (Bhagat et al., 2011; Zhang et al., 2018), link prediction (Kumar et al., 2020) and graph classification (Kriege et al., 2020; Cai et al., 2018).

Although the aforementioned architectures have demonstrated great success, recent research has shown considerable limitations of GNNs. First, more complex architectures tend to be computationally demanding. For instance, Graph Attention Networks (GATs) (Veličković et al., 2017) whose neighborhood aggregation mechanism employ computations of self-attentions to assign weights for neighboring nodes. Second, this methodology of neighborhood aggregation leads to an exponentially growing amount of information originating from exponentially growing neighborhoods that needs to be encoded within fixed-size node representation vectors, a phenomenon referred as *over-squashing* (Alon & Yahav, 2020). Furthermore, the varying number of nodes participating in each node's neighborhood leads to a highly varying amount of information that needs to be encoded within a fixed-length code, a phenomenon that we call the *varying neighborhoods*. Note that many other architectures (e.g., CNNs, MLPs, and RNNs) do not encounter this since they only accept fixed-size inputs or produce outputs with varying sizes that correspond to the input size. GNNs break this correspondence due to their neighborhood aggregation methodology. These phenomena become very prominent as the depth of the neighborhood grows, which limits our ability to develop

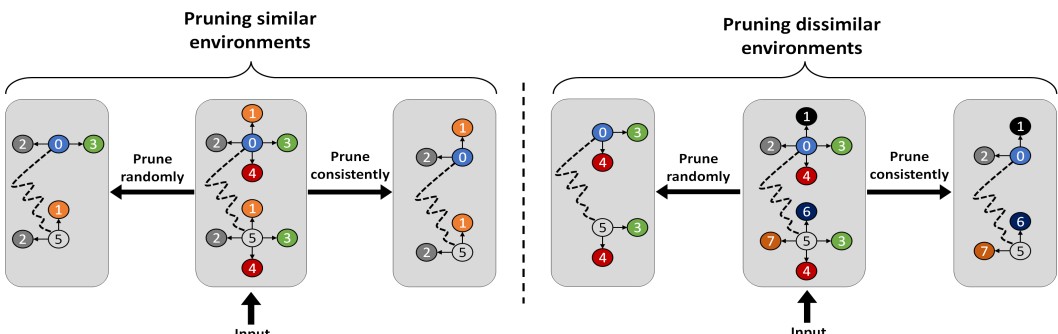

Figure 1: Illustration of consistent pruning versus non-consistent pruning. On the left, the input consists of a graph with two highlighted environments whose topology is similar because the central nodes have the same set of neighbors. In this case, consistent pruning results with a graph that preserves the similarity between these environments while non-consistent pruning, e.g., random removal of edges, results with dissimilar environments. On the right, the two highlighted environments have dissimilar topologies because the central nodes have different sets of neighbors. Consistent pruning is likely to preserve this dissimilarity, while *random* might bring the similarity between them closer.

deep GNNs. We further discuss these phenomena and provide explanations on how pruning edges could help mitigating them in Section 4.4.2.

A popular approach for tackling these problems is to apply some transformation or augmentation to the input graph. One prominent approach is Graph Sparsification (Hu & Lau, 2013; Spielman & Teng, 2008; Spielman & Srivastava, 2008; Calandriello et al., 2018) in order to accelerate GNNs by removing nodes and edges from a graph under the constraint of preserving the predictive performance (Rong et al., 2019; Srinivasa et al., 2020; Ye & Ji, 2021; Hamilton et al., 2017). In fact, Faber et al. (2021) showed that very often a significant amount of edges can be removed without degrading the performance of a model, which raises a concern regarding the usage of popularly used benchmarks. They claimed that those tasks can be virtually solved through node features only, implying that the graph topology has limited contribution and edges can be safely disregarded.

Motivated by the above discussion, we argue that graph sparsification can greatly improve the performance of GNNs in tasks where the graph topology is significant. Moreover, we argue that sparsification that is based on local properties of a graph is more in-line with the prevailing methodology of GCNs, as opposed to approaches that rely on global topologies, such as spectral methods. Therefore, we introduce Locality Sensitive Pruning, a new algorithm for edges pruning based on locality sensitive hashing (LSH) (Shakhnarovich et al., 2008). As a result, pruned graphs are qualified with structure dynamics that alleviate the ability to distinguish between different graphs and mitigate the computation burden. More importantly, similar environments of the input graph result in similar environments in the sparsified graph while dissimilar environments result in dissimilar environments in the sparsified graph with high probability, as depicted in Figure 1. Consequently, we preserve the ability to distinguish between environments of the graph that were distinguishable prior to the sparsification process.

## 2 RELATED WORK

**Graph Convolutional Networks (GCNs)** were introduced by Kipf & Welling (2016) and has been widely adopted for solving tasks involving graphs, such as node classification (Bhagat et al., 2011; Zhang et al., 2018), link prediction (Kumar et al., 2020), graph classification (Kriege et al., 2020; Cai et al., 2018), and more. The most basic form uses simple aggregation functions to obtain a node representation as a function of its neighbors, such as average and summation. Later, the model was extended to more complex architectures which introduce more sophisticated aggregation functions. For instance, Graph attention Networks (GATs) (Veličković et al., 2017) use dot-products based attention to calculate weights for each edge. Additionally, Corso et al. (2020) argued that

the aggregation layers are unable to extract enough information from the nodes' neighbourhoods in a single layer, resulting in a limited expressive power and learning abilities. To address this, they proposed Principal Neighbourhood Aggregation for Graph Nets (PNA) which combines multiple aggregators to improve the performance of the graph neural networks.

**Graph Sparsification**   aims to approximate a graph on the same set of vertices and edges (Benczur & Karger, 2002). Much research has been done in this field and several algorithms were proposed, including spectral sparsifiers (Spielman & Teng, 2008; Calandriello et al., 2018; Chu et al., 2020), sampling via Metropolis algorithms (Hübler et al., 2008), and others (Sadhanala et al., 2016). In the context of training graph neural networks, the motivation is twofold: 1. accelerate training by performing fewer message-passing operations, and 2. regularization. For the former, Srinivasa et al. (2020) propose FastGAT, a spectral sparsifier based on effective-resistance (Spielman & Srivastava, 2008) for acceleration of GATs. Specifically, they aim to accelerate GATs by constructing a sampled graph with far fewer edges for each attention head within the network. Chen et al. (2018) proposed FastGCN which samples vertices and edges through importance sampling and thus reduces the graph size. Another line of works (Rong et al., 2019; Chen et al., 2020; Hasanzadeh et al., 2020; Zheng et al., 2020; Luo et al., 2021; Kim & Oh, 2020; Ye & Ji, 2021) perform graph sparsification to achieve regularization, with the aim to prevent over-smoothing and overfitting (Li et al., 2018).

**Locality-sensitive hashing (LSH)**   is a widely used technique for finding pairs of similar items in a large set efficiently. It has been successfully applied for accelerating the Transformer model (Vaswani et al., 2017) by mitigating the burden of computing pair-wise attentions as demonstrated by the Reformer (Kitaev et al., 2020). This is done by approximation of attention computations based on locality-sensitive hashing, which replaces the $\mathcal{O}(L^2)$ factor in attention layers with $\mathcal{O}(L \cdot logL)$.

## 3   PRELIMINARIES

### 3.1   GRAPH CONVOLUTIONAL NETWORKS (GCNS)

A graph is a 2-tuple $G = (V, E)$ where $V$ is a set of nodes and $E \subseteq V \times V$ is a set of undirected edges connecting pairs of nodes. We consider the setting in which each node $v$ is associated with a feature vector $x_v \in \mathbb{R}^d$. The adjacency matrix of $G$, denoted by $A \in \{0, 1\}^{|V| \times |V|}$, associates each edge $e_{i,j} = (v_i, v_j) \in E$ with an entry $A_{i,j}$ indicating that $e_{i,j} \in E$. The number of edges connected to a node $v_i$ (also known as the degree of $v_i$) is denoted by $d_i = \sum_j A_{i,j}$. We define $D$ as a diagonal matrix where $D_{i,i} = d_i$ and 0 elsewhere. For what follows, we consider the Graph Convolutional Network (GCN) model from Kipf & Welling (2016). This model is recursively defined as

$$H^{(l+1)} = \sigma(\tilde{D}^{-\frac{1}{2}} \tilde{A} \tilde{D}^{-\frac{1}{2}} H^{(l)} W^{(l)}), \tag{1}$$

where $\tilde{A} = A + I$ is the adjacency matrix of the undirected graph $G$ with additional self-loops, obtained by adding the identity matrix $I$, $W^{(l)}$ is a weight matrix associated with the $l^{th}$ layer of the model, and $\sigma(\cdot)$ is some non-linear activation function. The notation $H^{(l)}$ relates to the result of the $l^{th}$ layer of the model, which is also the input to the $l + 1$ layer. The input to the first layer is $H^{(0)} = [x_1, \ldots, x_{|V|}]^{\mathsf{T}} \in \mathbb{R}^{|V| \times d}$. We intend to accelerate the training and inference of this model by reducing the input size.

### 3.2   LOCALITY SENSITIVE HASHING (LSH)

The Locality Sensitive Hashing algorithm is designed to approximate distances between vectors in a high-dimensional space in a lower dimensional space. It is widely applicable in fields like data mining and machine learning, and is particularly useful for tasks such as approximate nearest neighbors. This algorithm relies on the existence of a family $\mathcal{H}$ of hash functions mapping $\mathbb{R}^d$ to some universe $U$. The definition of LSH is as follows.

**Definition 1** *A family of hash functions $\mathcal{H}$ is called $(R, c, P_1, P_2)$-sensitive if for any $p, q \in \mathbb{R}^d$*

- $d(p, q) \leq R \Rightarrow P_{h \in_R \mathcal{H}}[h(q) = h(p)] \geq P_1$,

- $d(p, q) \geq cR \Rightarrow P_{h \in_R \mathcal{H}}[h(q) = h(p)] \leq P_2$.

An interesting hash functions family satisfies the constraint that $P_1 > P_2$. In our work, we utilize LSH by mapping feature vectors of graph edges into integers, namely "buckets", so that similar features are mapped into the same buckets. Then, the hash of a set of edges, consisting of edges in a neighborhood of a node, is the bucket with the minimal hash value, which is referred as *MinHash*. When applied to numerous sets, this process results with the same minimal buckets for similar edge sets. Consequently, by choosing to keep edges that correspond to the *MinHash* values, we aim to preserve the similarities of different graph regions.

## 4 LOCALITY SENSITIVE PRUNING

This section introduces the methodology of Locality Sensitive Pruning (LSP). Additionally, we provide a complexity analysis of the algorithm and address its limitations. Finally, we discuss the advantages of pruning edges from graphs and the prominent advantages of LSP over other methods.

### 4.1 OVERVIEW

Our aim is to modify the topology of a graph so that similar regions result in a similar, yet consistently sparsified structure. For this purpose, we base our approach on locality sensitive hashing that assigns signatures to edges so that similar edges are assigned with the same signature. Then, we specify *MinHash* as a rule for choosing which edges to keep; that is, the edges that make up the sparsified graph. In *MinHash*, the selected edges correspond to signatures that are mapped to minimal hash values using pseudo-random hash functions. These psuedo-random hash functions are used across all graph regions, and thus similar regions of the original graph result in similar regions in the resulting sparsified graph.

The procedure of sparsifying a graph using LSP is as follows. Given a graph $G = (V, E)$, LSP constructs a new graph $G' = (V, E')$ where $E' \subset E$. We assume every edge $(i, j) \in E$ is associated with an attribute $e_{i,j} \in \mathbb{R}^d$, and the set of $k$ pseudo-random functions $\mathcal{H} = \{h_z | \forall z \in [k] : h_z : \mathbb{R}^d \to \mathbb{R}\}$ maps $d$-dimensional vectors into scalars under the locality-sensitive-hashing criteria (see Section 4.2 for more details on the choice of these functions). For each node $u \in V$, we compute the hash values for all the edges connecting $u$ with its neighbors $\mathcal{N}_u = \{v \in V | (u, v) \in E\}$. Then, we base our choice of edges using the *MinHash* rule, that is, edges whose hashes are mapped to the minimal value among all the other edges. The overview of our sparsification method is described in Algorithm 1.

---

**Algorithm 1** Locality Sensitive Pruning

> **Input:** $G = (V, E)$                           ▷ The input graph
>          $\mathcal{H} = \{h_1, ..., h_k\}$    ▷ A set of $k$ locality-sensitive hashing (LSH) functions mapping edge
> attributes to scalars
> **Output:** A sparsified graph $G' = (V, E')$
> $E' \leftarrow \emptyset$
> **for** $u \in V$ **do**
>      $E_u \leftarrow \emptyset$
>      **for** $i \in [k]$ **do**
>          $v_{min} \leftarrow \arg \min_{v \in \mathcal{N}_u} h_i(e_{u,v})$                    ▷ Apply *MinHash*
>          $E_u \leftarrow E_u \cup \{(u, v_{min})\}$
>      **end for**
>      $E' \leftarrow E' \cup E_u$
> **end for**
> **return** $G' = (V, E')$

---

Note that LSP assumes the existence of edge attributes. However, many real-world problems incorporate only node attributes. In these cases, we construct edge attributes from the nodes connected to its ends, as we describe in Appendix A.

## 4.2 THE CHOICE OF LSH FUNCTIONS

The description of LSP does not define a specific family of hash functions. This is because different datasets contain attributes with different nature. Consequently, this requires choosing appropriate hashing methodologies for each dataset. In the following, we present two common choices for hash functions that we use in our experiments (Section 5). We refer the readers to (Paulevé et al., 2010) for a review of other LSH function choices.

**Binary signatures**   A LSH function that uses binary signatures is a composition $h = h_2 \circ h_1$ of a binary signature mapping $h_1 : \mathbb{R}^d \to \{0,1\}^d$ and a hashing functions $h_2 : \{0,1\}^d \to \mathbb{N}$ that maps binary signatures to integers. Specifically, we define $h_1$ as a thresholding function which is associated with a random vector $w$ (with each $w_i \sim N(0,1)$) that determines the threshold for each entry of an input vector $x$. Formally, the thresholding is performed by comparing the entries of $x$ and $w$:

$$h_1(x)_i = \begin{cases} 1, & x_i > w_i \\ 0, & \text{otherwise} \end{cases}. \tag{2}$$

Then, $h_2$ maps the result to integers using the well-known message-digest algorithm (Rivest, 1992) and bin it into one of $m$ bins by finding the remainder of the division by $m$. In our experiments (which will be presented in Section 5), we refer to this method as $LSP - T$.

**Random Projections**   A LSH hashing function that uses random projections (Shakhnarovich et al., 2008) aims to project inputs so that similar vectors result in proximate projections, and their quantization result in the same values. Let $w$ be some random vector with each $w_i \sim N(0,1)$. This random projection divides input vectors into bins of length $l$ using the hash function defined by

$$h_{w,b}(x) = \left\lfloor \frac{\langle x, w \rangle + b}{l} \right\rfloor. \tag{3}$$

Here, $b$ is a random variable sampled from the uniform distribution $b \sim U[0,l]$. The inner-product $\langle x, w \rangle$ is the projected value of $x$ onto the direction $w$, which is then shifted by the offset value $b$ and quantized into $l$ length bins. In our experiments, we refer to this method as $LSP - P$.

## 4.3 ANALYSIS

**Complexity**   The procedure of LSP is performed as a preprocessing step prior to the whole training phase due to its determinism. The pruned graph does not change between training episodes and thus should be performed only once. Assuming the dimensionality of each edge attribute is $d$, the complexity of computing each hash value is $\mathcal{O}(d)$ given that the hash function families proposed in Section 4.2 are used (note that this statement is valid for other common hash function families). Each of the $k$ hash functions computers $|E|$ hash values, one for each edge. In total, the running time complexity of LSP is $\mathcal{O}(k|E|d)$. The space complexity is $\mathcal{O}(1)$, because this only requires iterating over edges and keeping the minimal hash value, which requires a constant amount of memory.

**Limitations**   We base LSP on the assumption that the topology of the input graph plays a significant role in the learning procedure. For this reason, in problems where the graph topology is less meaningful, we do not expect a significant advantage of pruning using LSP over its rival, such as random removal of edges.

## 4.4 BENEFITS OF USING LSP

In the following, we present the beneficial reasons for pruning edges. While many claims are valid in general, that is, they are applied to other pruning methods other than LSP, we begin with advantages that are particular for LSP.

### 4.4.1 ADVANTAGES OF PRUNING USING LOCAL ENVIRONMENTAL CHARACTERISTICS

Several graph-related tasks, including node-classification among them, rely on the composition of the neighborhoods. This results from the core methodology of GCNs which aggregates information

from a node's local environment. Consequently, we want our pruned graph to preserve local properties of the original graph. For demystifying the necessity to specifically consider local properties for pruning, we exemplify a node classification scenario in which consistent pruning that preserves similarities between neighborhoods is necessary for successfully predicting the node classes.

**Example 2** *This example is illustrated in Figure 2. Consider a node classification scenario consisting of 2 classes. The criterion for a node to belong to class 1 is being a neighbor of both nodes $1, 4 \in V$. Similarly, the criterion for a node to belong to class 2 is being a neighbor of both nodes $2, 3 \in V$. Notice that for each environment on the left, there's a corresponding environment on the right which has at least 2 common neighbors. For example, the top left environment contains nodes $8$ and $20$ as neighbors of $F$, while on the right we have nodes $8$ and $20$ as neighbors of $B$.*

*Suppose that we want to reduce the degree of every node by $50\%$. When performing consistent pruning by choosing to keep the nodes corresponding to minimal identity (in our algorithm, these are the hash values) we encounter the following situation: on the left, we are left with environments consisting of nodes $E, F, G, H$, each of which has two neighbors $1$ and $4$. On the right, we are left with environments consisting of nodes $A, B, C, D$, each of which has two neighbors $2$ and $3$. These results will enable us to train a classifier that distinguish between the depicted components.*

*Meanwhile, suppose that we randomly prune $50\%$ of the edges of each node. Since there are $\binom{4}{2} = 6$ possibilities, the probability that we are left with environments consisting of nodes $8$ and $20$ both on the left and the right is $\left(\frac{1}{6}\right)^2 = \frac{1}{36}$. Consequently, a classifier trained on randomly pruned graphs using the described settings would have an error rate of $2.7\%$.*

**Class 1** $= \{v \in V | 1, 4 \in \mathcal{N}_v\}$   **Class 2** $= \{v \in V | 2, 3 \in \mathcal{N}_v\}$

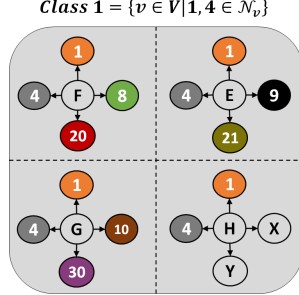   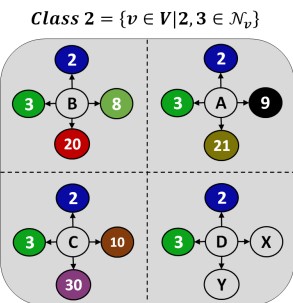

Figure 2: Example scenario in which we are tasked to classify nodes into 2 classes. Each of the nodes $A, B, C, D$ belong to class 2, while each of the nodes $E, F, G, H$ belong to class 1. Moreover, For each node among $\{E, F, H, G\}$ there exists a node among $\{A, B, C, D\}$ that the intersection of the sets of their neighbors has size of at least 2.

### 4.4.2 OTHER ADVANTAGES OF PRUNING EDGES OF A GRAPH

**Accelerating GCN training and inference** The outcome of pruning edges from a graph is an acceleration of each training iteration. To see this, recall the recursive definition of GCN (Equation 1) from Kipf & Welling (2016). In this definition, the weights matrix $W$ is of size $C \times F$ where $C$ and $F$ denote the number of input channels and filters respectively. Additionally, the matrix $D$ is a diagonal matrix of size $|V| \times |V|$ with at most $|V|$ non-zero elements, and the matrix $A$ is of size $|V| \times |V|$ whose number of non-zero elements is at most $|E|$. Apparently, the complexity of this matrices multiplication does not depend on the number of edges, however from Kipf & Welling (2016) we know that in practice $\tilde{A}H$ is efficiently implemented as a product of a sparse matrix with a dense matrix whose complexity is linear in the number of edges $\mathcal{O}(|E|FC)$. For GCNs that uses an attention mechanism, from Veličković et al. (2017) we know that the time complexity of a single attention head is $\mathcal{O}(|V|CF + |E|F)$. With this, we claim that pruning edges from an input graph linearly accelerates GCNs and their variants.

**Mitigating Over-Squashing** LSP simplifies the graph structure by removing edges. In turn, it reduces the amount of information needed to be encoded within the node representation vectors. Particularly, pruning of edges helps mitigating the *over-squashing* phenomenon in which information from the exponentially-growing receptive field is compressed into fixed-length node vectors

(Alon & Yahav, 2020). Since removing edges reduces the degree of each node, it also reduces its receptive field exponentially, thus preventing this destructive phenomenon.

We further discuss the *Neighborhood Variance* issue in Appendix C.

## 5 EXPERIMENTS

We validated LSP on 3 graph-related tasks: (1) Node classification; (2) Graph regression, and (3) Graph classification. The objectives of this empirical study were twofold:

- Assess the quality of the pruned graphs in terms of performance metrics. We train and test several models on various datasets and configurations, and measure the achieved performance metric identified with each dataset as a dependent of the pruning configurations.
- As we target acceleration of GNNs, we assess the amount of acceleration obtained by training and testing with pruned graphs. This is done by measuring the relative time required for training and testing on graphs that are pruned with various pruning configurations.

For all the experiments discussed in this section, for each dataset we choose a commonly used model that is identified with this dataset. For example, we use the GAT (Veličković et al., 2017) architecture proposed for the PPI (Zitnik & Leskovec, 2017) dataset because it achieves state-of-the-art performance at the time of writing this paper. We refer the readers to Appendix B.3 for more information about the models that we use for experimenting with each dataset. Additional implementation details are given in Appendix B.1.

**Evaluation Protocol** To evaluate the quality of the pruned graphs, we train and test models with various pruning configurations and evaluate their performance in terms of the metric identified with each dataset. Additionally, we measure the running times in order to demonstrate the relative acceleration achieved via pruning. These results are presented as the relative time required for each iteration compared to the model trained on the unpruned graph. It is important to note the observation that given a pruning ratio, the choice of pruning methodology does not impact the running times, thus we present a single curve that describes the acceleration for all pruning methods.

**Baselines** We compare the performance of 2 LSP variants (see Section 4.2) with *Random*, that relates to the method which randomly removes edges from a graph with a certain probability $p$. We do not compare LSP with methods that sparsify the input graph during inference, such as FastGCN (Chen et al., 2018) and DropEdge (Rong et al., 2019), which are discussed in Section 2. This is because LSP does not modify the training and inference phases of GNNs, as opposed to the aforementioned methods. Additionally, in order to demonstrate the advantage of using the combination of LSP with complex models over using simpler GNNs which are computationally cheaper in terms of acceleration, we compare the performance of these models with various pruning configurations against a basic GCN that is naturally computationally cheaper (described in Appendix B.3).

### 5.1 NODE CLASSIFICATION BENCHMARKS

**Data** We experimented with various publicly available datasets: GitHub (Rozemberczki et al., 2019), Cora (Bojchevski & Günnemann, 2017), CiteSeer (Bojchevski & Günnemann, 2017), PubMed (Bojchevski & Günnemann, 2017) and PPI (Zitnik & Leskovec, 2017). Detailed descriptions and statistics of these datasets are presented in Appendix B.2.

**Results** Results for node classification benchmarks are shown in Figure 3. Additional results are provided in Appendix C.3. The superiority of LSP is demonstrated through the experiments conducted on Cora and PPI. For these benchmarks, the predictive performance drops significantly for aggressively pruned graphs via random, while it is noticeable that LSP better preserves the accuracy of the model trained on the original graphs. Moreover, the basic models used for these experiments demonstrate a significant decrease in performance while the acceleration achieved by their simplicity is negligible. Notice that for PubMed and CiteSeer, the accuracy ranges are narrow. For instance, the model trained on PubMed using 100% of the edges achieves an accuracy score of 0.882, while the same model that is trained with 30% of the edges using random pruning achieves an

accuracy score of 0.859. Considering this relatively negligible difference, we observe that random pruning performs well and consistent pruning via either LSP-P or LSP-T preserves this performance. Additionally, for this experiment the basic model demonstrates performance that is comparable to the original model while being substantially faster. The same holds for CiteSeer.

To that end, we conclude that in cases where random pruning performs well, the graph topology has a negligible impact on the performance of the model. In these cases LSP performs marginally better than random. The superiority of LSP is well-demonstrated in benchmarks where the topology of the graph is impactful. In such cases, the performance of the models reduces significantly via random pruning, while pruning via an LSP variant preserves the local properties of the graph. To conclude this experiments series, we observe that the achieved inference acceleration is linear in the number of preserved, a result that matches our expectation as per our complexity expectations.

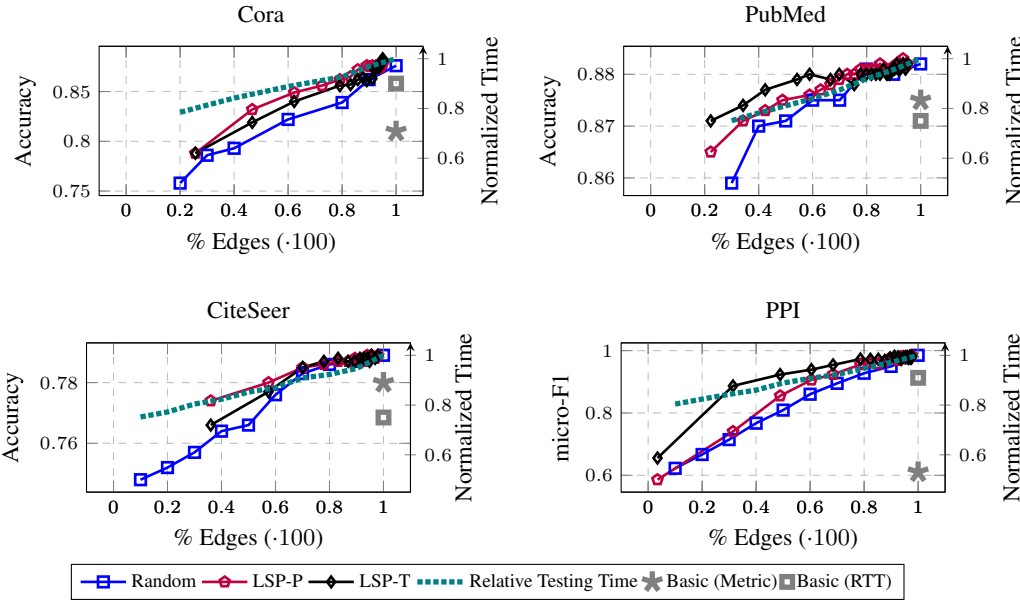

Figure 3: Test results for node classification problems. For each dataset, we report the results in terms of the metric identified with that dataset (in solid lines), and the running time per iteration (in densely dotted lines) under the specified pruning configurations.

## 5.2 GRAPH REGRESSION BENCHMARKS

**Data**   We experimented with datasets from the chemical world with the goal to predict chemical properties of molecules: QM9 (Wu et al., 2017) and ZINC (Sterling & Irwin, 2015). In both datasets, each graph is a molecule, consisting of nodes and edges that represent atoms and different types of bonds between the atoms respectively. The goal is to regress each graph to its properties, quantum properties in QM9 and constrained solubility (a synthetic computed property) in ZINC. Refer to Appendix B.2 for more details of these datasets.

**Results**   Results for the graph regression tasks are shown in Figure 4. We include additional results for other QM9 targets in Appendix C.3. Continuing the conclusions from the previous section, we observe that pruning via LSP perform better in terms of lower Mean Absolute Error (MAE) for both graph regression benchmarks.

## 5.3 GRAPH CLASSIFICATION BENCHMARKS

**Data**   In order to validate the effectiveness of LSP in extreme scenarios, we developed a generator for synthetic graph classification datasets. We provide the full list of parameters and their default values of this generator in Appendix B.2.1. We refer to the default generator configurations as those which we use in the following experiments.

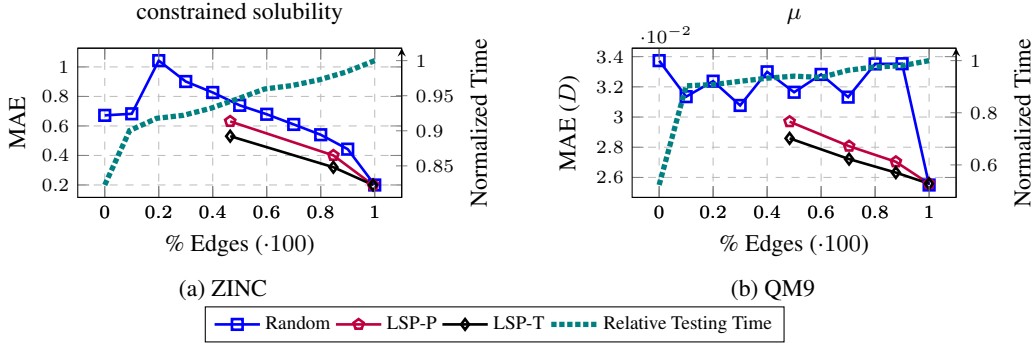

Figure 4: Mean Absolute Error (MAE) and normalized running times for graph regression problems. (4a) results for regressing the constrained solubility for molecules in the ZINC database. (4b) results for regressing the Dipole moment ($\mu$) for molecules in the QM9 database. Errors are presented as solid lines (Lower is better). Normalized running times are presented as dashed lines.

**Results** Results for graph classification on the synthetic benchmark are shown in Figure 5. This synthetic data generator enables us to simulate highly complex scenarios where the topology is impactful. This is well-demonstrated by the aggressive accuracy reduction when pruning with random. This yields the observation that LSP outperforms the random pruning method by a large margin, especially for mid-range pruning ratios. Moreover, as we use a highly complex model in this section (as described in Appendix B.3) that performs complex calculations for neighborhood aggregation, a significant acceleration is observed when compared to the model trained on the original

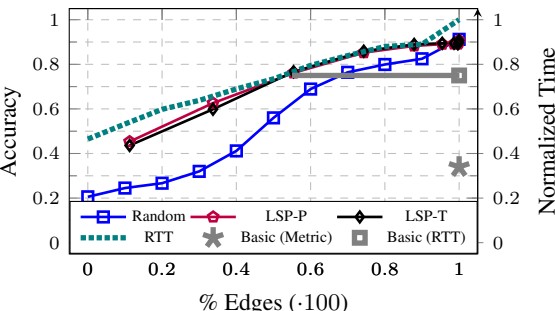

Figure 5: Accuracy and running time comparison for the synthetic graph classification dataset.

graphs. Additionally, this acceleration is depicted as a linear curve, which matches our expectations. The basic model used for this experiment (described in Appendix B.3.4) demonstrates a significant acceleration due to the lack of need to compute 16 attention heads per layer. This acceleration is also followed by a reduction of $\sim 54\%$ in accuracy: the original model trained on the original graphs achieves an accuracy score of $90.76\%$ while the basic model achieves an accuracy score of $34\%$. For the sake of comparison, the original model trained on pruned graphs with the configuration that achieves comparable acceleration results in accuracy score of $\sim 76\%$ for both LSP variants. In this configuration, nearly $45\%$ of the graph edges are pruned. From this result, we conclude that we were able to train a highly complex model that does not compensate for the performance while being accelerated via pruning its input.

## 6 CONCLUSIONS

We have presented Locality Sensitive Pruning (LSP), a systematic method for pruning graphs based on locality sensitive hashing. By relying on the local features of neighborhoods, LSP preserves local properties of the graph. Consequently, it does not impair the ability to distinguishing between samples that were distinguishable prior to the pruning process with high probability. We evaluated LSP through an extensive experimental study on 3 graph-related tasks. For these experiments, we trained several ad-hoc graph neural networks on diverse real-world datasets that went through the pruning process of 2 LSP variants. In all experiments, LSP demonstrates superior performance when compared to the baselines. At the same time, the reduction in the number of edges translates to a significant acceleration of the used models, which is linear in the number of edges. A possibly rewarding avenue of future research is the design of dedicated locality sensitive hash functions that capture informative features of the graph.

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

## A   CONSTRUCTION OF EDGE ATTRIBUTES

As discussed in Section 4, LSP assumes the existence of edge attributes. However, many graph datasets do not include such representation but include node attributes instead (or additionally). Additionally, some datasets have scalar values as edge and node representations (for instance, molecules with number of bonds as edge attributes and atomic number as node attribute). Those representations cannot be used as inputs to our hash functions because they expect vectors as input rather than scalars. Here we describe our choice of representation in each case.

Recall the definition of edge attributes from Section 4 - an attribute of an edge $(u,v) \in E$ is a $d$-dimensional vector denoted as $e_{u,v} \in \mathbb{R}^d$. Similarly, an attribute of a node $u \in V$ is a $q$-dimensional vector. For convenience, we use the name of a node itself as a notation for its attribute. We denote the final representation of an edge attribute which serves as an input to the $LSP$ algorithm as $e_{u,v}^{LSP}$.

**Node attributes only -** $e_{u,v}^{LSP} = [u,v]$.

**Both node and edge attributes -** $e_{u,v}^{LSP} = [u, e_{u,v}, v]$.

**Scalar node/edge attributes -** Let $r \in \mathbb{N}$ be an attribute of a node or edge (for example, atomic number or number of bonds between atoms in a graphs that represents a molecule). Assume $r$ accepts $m$ possible values, $r \in \{0, ..., m-1\}$. We generate a random matrix $M \in \mathbb{R}^{m \times m}$ and use the columns of $M$ as embeddings of the corresponding component; that is, the $r^{th}$ column of $M$ is the corresponding feature vector of the graph component associated with the attribute $r$.

**Example 3** *We finalize this section with a complete example of constructing edge attributes from a graph $G = (V, E)$ representing a molecule. We assume every node $v \in V$ is associated with a natural scalar $r_v \in \{0, ..., m_V - 1\}$ and every edge $e \in E$ is associated with a natural scalar*

$r_e \in \{0, ..., m_E - 1\}$. *Following our explanation for generating embeddings for scalar attributes, we generate two random real matrices, $M_V \in \mathbb{R}^{m_V \times m_V}$ and $M_E \in \mathbb{R}^{m_E \times m_E}$. Then, the columns of each matrix are used as feature vectors of nodes and edges as follow. Let $M^i$ denote the $i^{th}$ column of a matrix $M$. Then, following our explanation for constructing edge representations as input for LSP, given two nodes $u, v \in V$ and an edge connecting the two nodes $e_{u,v}$, we obtain the following: $e_{u,v}^{LSP} = [M_V^u, M_E^{e_{u,v}}, M_V^v]$.*

# B    EXPERIMENTS

## B.1    DETAILS

**Implementation Details -**    All experiments were run on Ubuntu 18.04 machine with Intel i9-10920X CPU, 93GB of RAM and 2 GeForce RTX 3090 GPUs. Our implementation of LSP is in Python 3. To generate graphs and perform training of graph neural models, we use Pytorch-Geometric (Fey & Lenssen, 2019) and use Pytorch (Paszke et al., 2019). Moreover, we all datasets used in this section are retrieved from the collection supplied by Pytorch-Geometric.

## B.2    DATASETS

### B.2.1    SYNTHETIC GRAPH DATASET GENERATOR

Here, we introduce the synthetic graph classification dataset generator. Table 1 summarizes the parameters used for the synthetic data generation and their default values, which we use for our experiment described in Section 5.3.

Table 1: Parameters of the synthetic dataset generator. For each parameter introduced in this table, the default parameter denotes the value that we use for our experiments in Section 5.3.

| Parameter | Description | Default |
|---|---|---|
| Number of samples | The number of graph instances in the dataset | 20,000 |
| Number of classes | The number of classes in the dataset | 100 |
| Minimum nodes | Minimum number of nodes in each graph sample | 40 |
| Maximum nodes | Maximum number of nodes in each graph sample | 60 |
| Node dimension | Dimensionality of the node representation vectors | 10 |
| Edge dimension | Dimensionality of the edge representation vectors | 40 |
| Connectivity rate | Degree of each node | 0.2 |
| Node centers std | Scatter of the node representation vectors | 0.2 |
| Edge centers std | Scatter of the edge representation vectors | 0.2 |
| Node noise std | Amount of additive noise for node representation vectors | 0.25 |
| Edge noise std | Amount of additive noise for edge representation vectors | 0.1 |
| Is symmetric | Whether to enforce symmetry on each graph | False |
| Node removal probability | The probability of remove a node from a graph sample, after the graph is initally generated | 0.1 |

### B.2.2    REAL-WORLD DATASETS

Here, we describe the various real-world datasets used in this paper for different tasks.

**Node Classification**

- **GitHub** - A social network where nodes correspond to developers who have starred at least 10 repositories and edges to mutual follower relationships. Node features are location, starred repositories, employer and e-mail address. The task is to classify nodes as web or machine learning developers.

- **Cora, CiteSeer and PubMed** - These are citation network with labels of paper topic. Each node represents a publication corresponding to different class (seven classes for Cora, 6 classes for CiteSeer and 3 classes for PubMed). The citation network consists of links representing citations. In Cora and CiteSeer, each publication in the dataset is described by a 0/1-valued word vector indicating the absence/presence of the corresponding word from the dictionary. In PubMed, each publication in the dataset is described by a TF/IDF weighted word vector from a dictionary which consists of 500 unique words.

- **PPI** - This dataset is made up of 24 graphs, with each graph corresponding to a different human tissue. Each node represents a protein and is associated with features made up of motif gene sets and immunological signatures. The task is to classify each protein into gene ontology sets (121 in total).

Statistics for these datasets are shown in Table 2.

Table 2: Node Classification Datasets Statistics

| Dataset | Github | Cora | CiteSeer | PubMed | PPI |
|---|---|---|---|---|---|
| Number of nodes | 37700 | 2708 | 3327 | 19717 | 44906 |
| Number of edges | 578006 | 10556 | 9104 | 88648 | 1136460 |
| Number of classes | 2 | 7 | 6 | 3 | 121 |
| Average degree | 30 | 7 | 5 | 8 | 50 |

**Graph Regression**

- **QM9** - The Quantum Mechanics 9 database contains around 130k small organic molecules with up to 9 heavy atoms and their physical properties in equilibrium, computed using density functional theory calculations. Following previous works (Zhang et al., 2020; Klicpera et al., 2020), we randomly sample 110k molecules for training, 10k for validation and the rest for testing. Results are presented in terms of Mean Absolute Error (MAE).

- **ZINC** - A collection of chemical compounds prepared especially for virtual screening. In this collection, a graph represents a molecule, with node features indicating the atom type and edge features the type of chemical bond between two atoms. The goal is to regress the molecule's penalised water-octanol partition coefficient. We use the original train/validation/test splits from Gómez-Bombarelli et al. (2018) as provided in Pytorch-Geometric (Fey & Lenssen, 2019). In similar to Dwivedi et al. (2020), we randomly select 12K for efficiency.

## B.3 MODELS

As stated in Section 5, we use the state-of-the-art model identified with each dataset to compare performance of LSP with other baselines. In the following, we provide descriptions of these models. After introducing these models, we present the basic architectures used as baselines for our experiments.

### B.3.1 NODE CLASSIFICATION

- **GitHub** - GAT model used in the comparisons in (Rozemberczki et al., 2021). This model consists of 2 attention layers that aggregate information up to 2-hop neighbourhoods with 1 attention head. For non-linearities we used leaky rectified linear unit (Leaky ReLU) with negative slope of 0.2.

- **Cora**, **CiteSeer** and **PubMed** - We use the methodology introduced in GraphSage (Hamilton et al., 2017) in combination with GAT, which has 2 layers. Following GraphSage, we sample at most 10, 15 neighbors for the first and second hop respectively for training. For testing, we sample the whole neighborhood of a node. The first layer has 2 attention heads, where each attention result is has hidden size of 8 and concatenated to the others. The second layer, which is used for calculating the output has 1 attention head. We use exponential linear unit (ELU) as activation between the layers.

- **PPI** - GAT model with 3 layers from Vaswani et al. (2017). The hidden dimension of each layer is 256 and the number of attention heads in each layer is 4,4,6 for the first, second and third layer respectively. For non-linearities we used exponential linear unit (ELU).

### B.3.2 GRAPH REGRESSION

- **QM9** - For all experiments, we use the MXMNet architecture proposed in (Zhang et al., 2020), with best performing configurations presented in the original paper.
- **ZINC** - We use the PNA architecture (Corso et al., 2020) with its default settings.

### B.3.3 GRAPH CLASSIFICATION

- **Synthetic dataset** - We construct a GAT model with 3 layers with 16 attention heads each. The hidden size of the intermediate layers is 40. The first two layers concatenate the outputs of the attention heads for output vector of size $40 \cdot 16 = 640$. For non-linearities, we apply ReLU between every two consecutive layers. Then, in order to perform the final classification, we average the the result of the third attention layer and aggregate the obtained node representations using global mean pooling. The result is later fed into a dropout layer which cancels 50% of the activations. Finally, we feed the result into a linear layer which provides the final prediction of the model.

### B.3.4 BASIC MODELS

The basic models used as baselines demonstrate the time-performance trade-off via training computationally cheaper models that deliver less performance when compared to computationally extensive models trained on pruned graphs. As we use GATs (Veličković et al., 2017) in all experiments of node-classification and graph classification, the basic models for these experiments are created by removing the attention mechanism while keeping all the accompanied configurations as is. As a result, we obtain GCN (Kipf & Welling, 2016) variants variant of the original models that aggregate messages from neighboring nodes by averaging instead if weights computed via attention heads.

## C   REDUCING THE NEIGHBORHOODS VARIANCE

The depth of a node's neighborhood determines the maximal path length between a node and other nodes contained in its neighborhood. A neighborhood of depth $k$ is called a k-hop neighborhood. As we increase the depth of the neighborhoods, the number of nodes contained within the neighborhoods increases. However, not all neighborhoods increase in the same way, which in turn leads to a variable size of neighborhoods - a phenomenon we call *Neighborhood Variance*, as exemplified in Figure 6. For the two given nodes $v_1, v_2$, their 3-hop neighborhoods have different sizes, which in turn translates to a variable amount of information that has to be encoded within a fixed-length code.

To see this, we measure the amount of nodes reachable within $k$ hops for each node $v$ in a graph $G$, denoted as $\mathcal{N}_G^{k,v}$. We claim the following:

**Proposition 4 (Pruning edges for reducing prevents over-squashing)** *Let* $G = (V, E)$ *be a graph and consider a sparsified version* $G' = (V, E')$ *so that* $E' \subset E$. *For each* $v \in V$, *denote by* $\mathcal{N}_G^{k,v}$ *the set of nodes reachable from* $v$ *within* $k$ *hops in the graph* $G$. *Then, there exists* $k_{max} \in \mathbb{N}$ *so that for all* $k \leq k_{max}$ *the variance of the amount of neighboring nodes needed to be encoded into the representation vector of a node satisfies:*

$$Var(|\mathcal{N}_{G'}^{v,k}|) \leq Var(|\mathcal{N}_G^{v,k}|). \tag{4}$$

Suppose that a $k$-layer GCN (for $k \leq k_{max}$) is trained to solve a task by aggregating information from $k$-hop neighborhoods into node representations. Then, from Proposition 4 we know that the neighborhood size of each node in the pruned graph has less variance and thus a similar amount of information has to be encoded within the representation vector of each node. Consequently, oversquashing is avoided. To demonstrate the scope of this phenomenon, we provide statistics about neighborhood size variance from popular real-world datasets in Appendix C.1. Additionally, we justify Proposition 4 in Appendix 4.

### C.1 NEIGHBORHOOD VARIANCE IN REAL-WORLD GRAPHS

Figure 6 depicts this variance for various neighborhood depths and pruning settings for certain datasets. For all datasets, we observe an exponential upward trend in the variance of the neighborhood sizes as more edges are incorporated in the graph for certain values of neighborhood depths ($k$). Additionally, an increase in the depth of the neighborhood (higher $k$ values) leads to a significant increase in this variance, which explains the performance degradation of deep GCNs due to over-squashing. This verifies our conclusion from Proposition 4 for real-world scenarios.

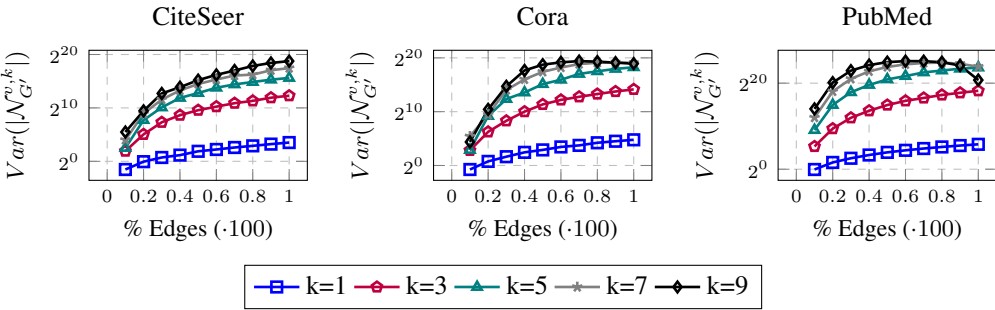

Figure 6: Variance of neighborhood sizes as a function of edge percentage preserved from the original graph (Log scale).

### C.2 JUSTIFICATION OF PROPOSITION 4

This claim is true for $k_{max} = 1$, thus proving its existence:

The size of the 1-hop neighborhood of $v$ equals the number of edges connected to it. Assuming the ratio of preserved edges in the pruned graph $G'$ is $0 \leq p < 1$, we have:

$$Var(|\mathcal{N}_{G'}^{v,1}|) = Var(d_{G'}(v))) = E[d_{G'}(v)^2] - E[d_{G'}(v)]^2 =$$
$$= E[(p \cdot d_G(v))^2] - E[p \cdot d_G(v)]^2 = E[p^2 \cdot d_G(v)^2] - E[p \cdot d_G(v)]^2 =$$
$$= p^2 \cdot E[d_G(v)^2] - p^2 \cdot E[d_G(v)]^2 = p^2 \cdot (E[d_G(v)^2] - E[d_G(v)]^2) =$$
$$= p^2 \cdot Var(d_G(v))) \leq Var(d_G(v))) = Var(|\mathcal{N}_G^{v,1}|)$$

### C.3 ADDITIONAL RESULTS

### C.4 NODE CLASSIFICATION

Additional results for node classification benchmarks are presented in Figure 7.

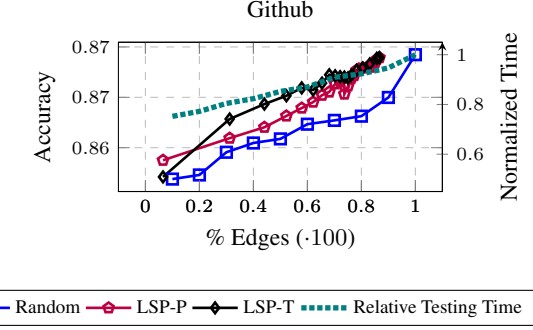

Figure 7: Additional results for node classification problems. We report the results in terms of the metric identified with the dataset (in solid lines), and the running time per iteration (in densely dotted lines) under the specified pruning configurations.

## C.5 GRAPH REGRESSION

Prediction results for all QM9 targets are presented in Figure 8.

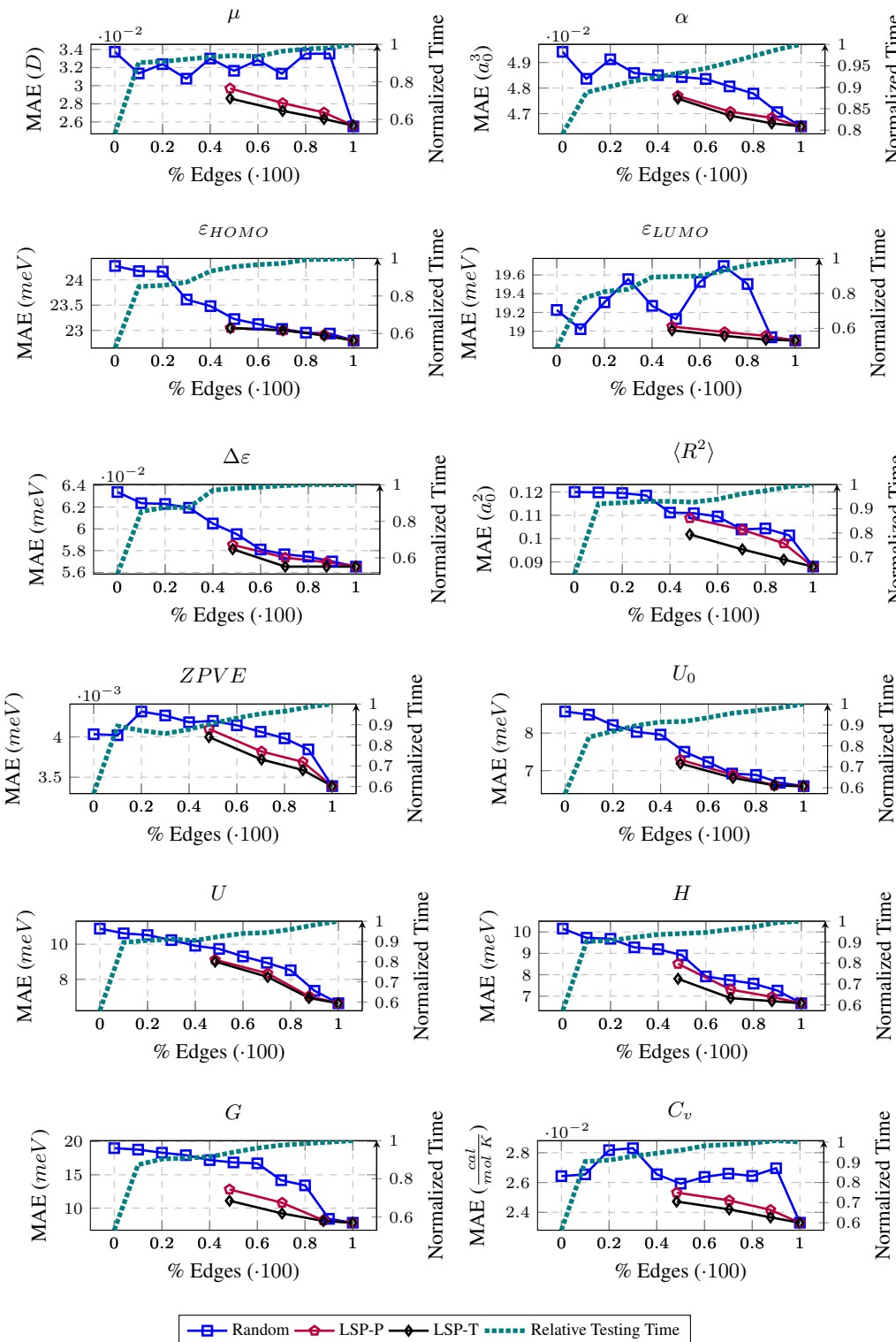

Figure 8: MAE and normalized running times for additional QM9 targets. Errors are presented as solid lines (Lower is better). Normalized running times are presented as dotted lines.

