# OpenReview forum: "LSP : Acceleration and Regularization of Graph Neural Networks via Locality Sensitive Pruning of Graphs"
_ICLR.cc/2022/Conference — ICLR 2022 Submitted_

### Official Review · Reviewer_7NLP · 2021-10-24

**Correctness:** 4
**Technical Novelty And Significance:** 2
**Empirical Novelty And Significance:** 3
**Recommendation:** 3
**Confidence:** 4

**Main Review:**

Sparsification of graph inputs is a valuable research direction as highlighted with the _over-squashing_ problem with GNNs. LSH is a popular approach for edge pruning in graph clustering and seems like a good choice here. Example 2 motivated the need for locality based reduction of edges, where a toy node classification example depends on 2 neighbours for correct classification, and showing that if each node has 4 edges, and are randomly reduced to 2, the error of misclassification is small; relevant when the topology of the graph is mainly important, over node attributes.

The experiments showed benefits of LSP over randomly removing edges, but no other methods were compared which I felt is missing. There are numerous other ways to prune (edge-prune) a graph, for example graphical lasso [1] and for efficient computation large scale precision matrix estimation [2], which can be applied to graph edge pruning [3] and there exists other related work not mentioned for sparsiyfing graphs in GNNS which could be directly compared for predictive performance of GNNs, for example [4,5], (with a wall-clock time to compare accuracy vs inference speed for example). Stronger motivation for LSH, over other methods of edge pruning, for example [2] (see [3] for example use of edge pruninig) and comparison with other edge-pruning methods would be beneficial here, and comparison with other GNN sparsification methods would strengthen the emprical evidence for the benefits of the proposed LSP.

1. https://en.wikipedia.org/wiki/Graphical_lasso
2. Zhang, Richard, Salar Fattahi, and Somayeh Sojoudi. "Large-scale sparse inverse covariance estimation via thresholding and max-det matrix completion." International Conference on Machine Learning. PMLR, 2018.
3. Strahl, Jonathan, et al. "Scalable probabilistic matrix factorization with graph-based priors." Proceedings of the AAAI Conference on Artificial Intelligence. Vol. 34. No. 04. 2020.
4. Wan, Guihong, and Harsha Kokel. "Graph Sparsification via Meta-Learning." DLG@ AAAI (2021).
5. Chiang, Wei-Lin, et al. "Cluster-gcn: An efficient algorithm for training deep and large graph convolutional networks." Proceedings of the 25th ACM SIGKDD International Conference on Knowledge Discovery & Data Mining. 2019.

**Summary Of The Paper:**

A pre-processing algorithm is presented to prune (sparsify) an input graph for a graph neural network (GNN), aimed at preserving the local neighbourhood structure. The algorithm uses locality sensitive hashing (LSH) on the graph edge embeddings, placing each edge into an integer 'bucket', the selection of which edges to preserve uses MinHash. Two variants of the algorithm are presented; one uses binary thresholding and the other uses random projections. This pruning is referred to as locality sensitive pruning (LSP).

Advantages of LSP are explained and experiments focus on the gained inference speed vs preserved prediction performance of a GNN model, when pruning the graph. As a baseline they prune the graphs randomly and show how their approach is better at preserving the prediction performance than this simple baseline. Experiments use GNNs for node classifcation, graph regression and graph classification on numerous datasets, where the a good performing GNN, typically graph attention networks (GATs), are used.


**Summary Of The Review:**

The main concern with the paper is the lack of comparison with any other methods that sparsify the graph. Firstly, the contribution here is a separate pre-processing step of the graph that uses edge pruning. There are numerous approaches to prune edges for example proabilistic methods that use the node attribute similarities to remove edges with negative correlations in the node attributes (a graphical lasso approach) and this shows to improve performance for then using the pruned graph as side information, in graph theory there are more methods for simply removing edges; It is not clear why LSH is a better choice. Secondly, there are other methods for sparsification of the graph for graph convolutional networks (GCNs) that not only sparsify but improve the performance. I think comparison with these existing GNN sparsification methods is important as the end goal of speeding up inference time while preserving predictive performance is the same.

With these comparisons, it will better position the paper. The area of sparsification of GNN is I believe an important one, so I feel this work has a lot of potential but there is already work on this that should be compared.

---

### Official Review · Reviewer_HFfr · 2021-11-01

**Correctness:** 3
**Technical Novelty And Significance:** 2
**Empirical Novelty And Significance:** 2
**Recommendation:** 3
**Confidence:** 4

**Main Review:**

**Pros**:
- Graph sparsification is an important and interesting problem due to the high computation complexity of GNNs.
- Based on the discussion in the related work section, the proposed LSP is novel for graph sparsification. Besides, the idea of using  Locality-Sensitive Hashing makes sense to me.
- The experiment section is well structured. It’s nice to see the authors evaluate the performance of the model on different underlying tasks.

**Cons**:
- **Novelty (main concern)**. Local Sensitive Hashing (LSH) has been used in Transformers to sparsify the fully connected self-attention computation graph in Transformer. LSP seems like a straightforward application from NLP to graph domain. From this paper, it is not clear why applying LSH to graph neural network is the challenge.
- **Motivation**. The author motivates the LSP using two limitations of existing GNN models in Section 1: (1) computation complexity and (2) varying neighborhoods (i.e., varying number of nodes participating in the node’s neighborhood leads to varying amounts of information to encode within a fixed-length code). It is not clear why varying neighborhoods will hurt the performance of GNN. To support their claim, the authors provide a discussion on the importance of reducing the variance of the neighborhood size in Appendix C, which is not convincing. In particular, in Appendix C, the authors argue that “by reducing the variance of the neighborhood size of each node can avoid over-squashing” which is not true because “over-squashing” is only relevant to the neighborhood size but irrelevant to the neighborhood size variance. On the other hand, the relative difference in neighborhood size can be important information to distinguish two nodes when using the forward propagation rule as defined in Eq. 1 since the aggregation weight is 1/\sqrt{deg(i)deg(j)}, which take the node degree information into considerations.
- **Experiment**. This paper lack enough comparison to baseline methods to indicate the effectiveness. The authors only compare with Random sampling-based sparsification methods: FastGCN and DropEdges. However, there are existing neural network-based deterministic graph pruning/sparsification methods that need to be compared. For example [1,2] as already cited by the authors and missing related works [3]. Besides, even for random-based methods, a stronger/more recent baseline such as neighbor sampling (e.g., GraphSAGE [4]), layerwise sampling (e.g., LADIES [5]), and subgraph sampling (e.g., GraphSAINT [6]) are expected to be compared since they are reported with better accuracy than FastGCN and DropEdges. More importantly, from the experiments section, LSP is still sacrificing performance a lot. For example, in Figure 3 (Cora), even just removing 20% edges, there is already around a 2% performance drop, the number continuously increasing as fewer edges are used. The same also happens to other datasets.
- **Clarification**. This paper is a Local Sensitive Hashing (LSH)-based sparsification algorithm, but LSH is not well introduced and is a little bit confusing. For example, in Section 3.2 Definition 1, the authors use notation d for both dimensions (e.g., p,q \in \mathbb{R}^d) and distance measure (e.g., d(p,q)). Besides, what are P1, P2? The authors are suggested to spend more effort on introducing the LSH algorithm before proceeding to their LSH-based algorithm.

[1] Learning to drop: Robust graph neural network via topological denoising.

[2] Fast graph attention networks using effective resistance-based graph sparsification.

[3] Learning Discrete Structures for Graph Neural Networks.

[4] Inductive Representation Learning on Large Graphs.

[5] Layer-Dependent Importance Sampling for Training Deep and Large Graph Convolutional Networks.

[6] GraphSAINT: Graph Sampling-Based Inductive Learning Method.

**Minor comments**:
- The proposed LSP is sparsifying the original graph by using the similarity between nodes. This requires an assumption on the homogeneity of the graph (i.e., nodes that are connected by edges are similar). What happens to other graph structures? For example in the user-item graph or heterogeneous graph (nodes that are connected by edges are dissimilar),  this type of similarity assumption does not hold.
- The model configuration in the experiment sections can be improved. For example, weight decay ratio, dropout, number of layers, learning rate, to help reproduce the results.


**Summary Of The Paper:**

 Training GNNs on a large graph requires significant computation resources. Besides, the noisy nature of the real-world graphs might cause GNNs to overfit. To overcome the above limitations, this paper proposes a graph pruning algorithm Locality-Sensitive Pruning (LSP) to sparsify the original graph into a sparse graph. Empirical results show that LSP can remove a significant amount of edges from the original large graph, reduce the computation complexity and training time, but with some compromise on the model performance.



**Summary Of The Review:**

Despite the pros summarized above, the paper lack enough novelty and requires more effort on experiments and clarification on several important aspects as mentioned above (e.g., clarification on the why neighborhood size variance and LSH).

---

### Official Review · Reviewer_Amda · 2021-11-01

**Correctness:** 2
**Technical Novelty And Significance:** 1
**Empirical Novelty And Significance:** 1
**Recommendation:** 1
**Confidence:** 4

**Main Review:**

Pro:
1. The paper is well written and clear for motivations. Easy to follow.

Cons:
1. The designed method is too simple: just a preprocessing and not interact with learning.
2. Baselines are limited. And result doesn't improve over non-sparsified version.
3. The author keep emphasizing that the proposed method can help: 1) training efficiently 2) regularization. However the result doesn't support them. First, there is no large dataset to show the benefit over training time. Second, regularization should help improving over non-sparsified version but it loses imformation and does worse than non-sparsified version.


**Summary Of The Paper:**

The paper proposes to prune graph structure based on hashing edge features with LSH. The proposed method is a deterministic preprocessing. The author evaluates the proposed method by comparing with other sparsification methods.

**Summary Of The Review:**

I vote for clear rejection. The proposed method doesn't show any strength in training large dataset and help on overfitting with more regularization.

---

> ### Author Response · Authors · 2021-11-19
> **Response to Amda**
>
> Thank you for your time spent for reviewing our paper. In what follows, we address specific comments.
>
> ```
> The designed method is too simple: just a preprocessing and not interact with learning.
> ```
> The effectiveness of the method is our goal. While it is also claimed to be very simple above, considering simplicity as a disadvantage is somewhat surprising. The method was developed to tackle an inherent problem regardless of its simplicity. In retrospect, it makes it easier to implement in various frameworks, which could be considered as an advantage.
>
> ```
> Baselines are limited. And result doesn't improve over non-sparsified version.
> ```
> We have demonstrated the effectiveness of LSP on various tasks that involve many datasets. Some datasets are indeed very small, and the advantage of the LSP over *Random* is not noticeable. It is important to include these experiments to show that *LSP* does not harm the performance in these cases. The advantage of *LSP* is well demonstrated on PPI and the synthetic datasets. On PPI,  *LSP-T* was able to preserve nearly 90% of the performance while pruning more than 50% of the edges. This is turn leads to a noticeable acceleration. Meanwhile,  the performance drops by nearly 30% when using *Random*, as demonstrated in Figure 3.
>
> The advantages are also demonstrated in the experiments on the synthetic dataset, where the acceleration is even more noticeable due to the computationally demanding architecture used in this experiment.

---

### Official Review · Reviewer_bzRT · 2021-11-02

**Correctness:** 2
**Technical Novelty And Significance:** 2
**Empirical Novelty And Significance:** 2
**Recommendation:** 3
**Confidence:** 4

**Main Review:**

he problem studies in this paper is an important research direction and a proper solution to the problem can be quite valuable. However, I have the following reservations about the current work.

[Experimental Results] For a paper that aims at accelerating GNNs, it is expected to provide results on larger-sized datasets. This is especially the case because as the authors mention, in 2 out of the 4 datasets in Figure 3 a random selection strategy seems to be on-par with the proposed approach. Why not use larger graphs from, e.g., the OGB? Also, why do the curves for the proposed approaches stop at around 0.45 in Fig 4?

[Baselines] While there are several existing works that solve a similar problem as that of the current paper (as mentioned in the main text), the proposed approach has been compared only against a random baseline which is not convincing.

[Intuition] One thing that is not clear to me after reading the paper is why selecting similar edges is a good pruning strategy. Example 2 of the paper is also too artificial and does not help much (with a slight modification of Fig 2, one can easily construct a counterexample where random selection has a 1/36 chance of selecting the right pair of points, whereas the proposed approach has a 0 chance).

[Correctness of MinHash] It seems to me that Algorithm 1 and the hash functions in section 4.2 do not correspond to MinHashing (I might be missing something so this could be resolved during rebuttal). Here’s what I understand from the description.
There are k hashing functions h_1, … h_k.
For each node v, for each hash function h_i, the neighbour that minimizes h_i is added to the neighbours of v.
Looking at Equation (3), the neighbouring node that minimizes h_i is the one minimizing floor((<x, w_i> + b_i) / l).
The neighbour that minimizes the above value is the one that is farthest away from w (assuming l is positive).
So the algorithm ends up selecting k neighbours that are farthest away from the random vectors, which does not seem like MinHashing.

**Summary Of The Paper:**

For large-size graph structures, the neighborhood aggregation operation of GNNs makes their training and inference slow and expensive. To remedy this problem, one solution proposed in the literature is to increase the sparsity of the graph by selecting a subset of the edges and only running the neighborhood aggregation over those edges. This paper develops an LSH-based approach for edge selection which can be run as a pre-processing step.

**Summary Of The Review:**

This paper studies an important problem, but the proposed solution is not well-motivated and the experimental results are inconclusive.

---

> ### Author Response · Authors · 2021-11-20
> **Response to bzRT**
>
> Thank you for your time spent for reviewing our paper. In what follows, we address specific comments.
>
> ```
> Also, why do the curves for the proposed approaches stop at around 0.45 in Fig 4?
> ```
> The average node degree of these graph is small. LSP prunes the graph in a local manner and considers every node individually. Consequently, the pruning ratios for nodes with very few neighbors grow rapidly. As the number of hash functions grows, more edges are restored and the degree of nodes with very few neighbors begins to saturate, while edges of nodes with many neighbors can still be added.
>
> ```
> One thing that is not clear to me after reading the paper is why selecting similar edges is a good pruning strategy
> ```
> The motivation for preserving similarities and dissimilarities of neighborhoods stems from the methodology of GCNs, which aggregate local information from the neighborhood of each node. This information is the only component on which the prediction depends, and it gives the ability to discriminate between different entities. However, pruning of edges distorts the structure and the neighborhoods and causes a loss of information. In order to preserve the discriminative power of GCNs, we assume that the similarities of neighborhoods should be preserved. This property is barely preserved by random pruning of edges. Also, the existence of this property was empirically demonstrated by the experimental results on various datasets.
>
> ```
> It seems to me that Algorithm 1 and the hash functions in section 4.2 do not correspond to MinHashing (I might be missing something so this could be resolved during rebuttal). Here’s what I understand from the description. There are k hashing functions h_1, … h_k. For each node v, for each hash function h_i, the neighbour that minimizes h_i is added to the neighbours of v. Looking at Equation (3), the neighbouring node that minimizes h_i is the one minimizing floor((<x, w_i> + b_i) / l). The neighbour that minimizes the above value is the one that is farthest away from w (assuming l is positive). So the algorithm ends up selecting k neighbours that are farthest away from the random vectors, which does not seem like MinHashing.
> ```
> The interpretation is indeed correct. The original $MinHash$ algorithm estimates the similarities of sets, which are sets of edges in our case. If two environments are similar, the edges that participate in both sets would have similar properties, including how far they are from $w$ and their minimal hash value $h_i(v_{min})$.

---

### Official Review · Reviewer_WZGy · 2021-11-02

**Correctness:** 4
**Technical Novelty And Significance:** 2
**Empirical Novelty And Significance:** 4
**Recommendation:** 5
**Confidence:** 4

**Details Of Ethics Concerns:**

There is no foreseeable ethics concern with this paper.

**Main Review:**

Strong points:
S1. The problem is well motivated to alleviate the computational burden, over-squashing, and varying neighborhoods problems in GNN models.
S2. Combing the LSH technique with GNN models for graph pruning is an interesting attempt.
S3. The paper presents several empirical results to demonstrate the effectiveness of LSP.
S4. The paper presents many insightful examples to illustrate the superiority of LSH-based pruning over the random pruning method.

Weak points:
W1. The paper employs a canonical hashing technique, LSH, to prune graphs for GNN acceleration and regularization. Although this idea is interesting, the primary contribution of this paper is a combination of existing techniques rather than a new method, insightful observation, or theoretical exploration.

W2. As stated in the abstract, the paper is motivated by the increasing graph sizes and the computational costs of GNN models. However, the graphs used in the experiments are modest. Additionally, a line of literature devoted to the scalability of GNN is missing. For example:
[SIGN] SIGN: Scalable Inception Graph Neural Networks.
[GBP] Scalable Graph Neural Networks via Bidirectional Propagation.
[PPRGo] Scaling Graph Neural Networks with Approximate PageRank.

W3. The relative time of the baseline method, random pruning, is omitted from the experiments. In comparison to random pruning, how does LSP perform in terms of relative time and accuracy trade-offs?

W3. The presentation of the paper needs some improvements. Please refer to the minor comments for details.


Minor comments:
M1. Citations should be added to support the two arguments in the last paragraph of Sec 1.
M2. Page 6: Could you provide more explanations on the sentence: “Apparently, the complexity of this matrices multiplication does not depend on the number of edges, …”.
M3. Page 8: to that end -> in the end.

**Summary Of The Paper:**

This paper proposes LSP, a graph pruning method based on Locality-Sensitive Hashing (LSH) for GNN acceleration and regularization. The paper shows that the LSP method can preserve the edge similarity after the pruning process. Additionally, the computational cost of GNN is reduced due to the graph sparsification.

**Summary Of The Review:**

The paper's idea is interesting, as evidenced by the experimental results. However, the theoretical depth and novelty may not be enough to meet ICLR's standards. Additionally, the concerns raised in W2 and W3 should be addressed.

---

> ### Author Response · Authors · 2021-11-19
> **Response to WZGy**
>
> We sincerely thank you for your comprehensive comments on our paper. We address specific comments in the following.
>
>
> ```
> Although this idea is interesting, the primary contribution of this paper is a combination of existing techniques rather than a new method, insightful observation, or theoretical exploration.
> ```
> The purpose of the paper is to propose a pruning strategy that is in-line with the training and inference procedures of GCNs. Although LSH was previously applied to a vast amount of applications, we here demonstrate the possibility of utilizing this highly beneficial method for our application, and this is where the novelty lies. The paper demonstrates to the public that the combination of LSH for pruning graphs and GCNs is beneficial. This insight is nonetheless trivial, and worth the vast experiments.
>
> ```
> The relative time of the baseline method, random pruning, is omitted from the experiments. In comparison to random pruning, how does LSP perform in terms of relative time and accuracy trade-offs?
> ```
> As stated in Section 5, under the **evaluation protocol**:
> *It is important to note the observation that given a pruning ratio, the choice of pruning methodology does not impact the running times, thus we present a single curve that describes the acceleration for all pruning methods.*
>
> The presented relative acceleration refers to all pruning methods.
>
> ```
> Could you provide more explanations on the sentence: “Apparently, the complexity of this matrices multiplication does not depend on the number of edges, …”.
> ```
> The size of the adjacency matrix $A$ of a given graph is fixed and does not depend on the number of edges. Removing edges from a graph would not impact the size of $A$. The multiplication of $AH$ in Eq. 1 is directly affected by the sizes of the two matrices. However, since $A$ is known to be a sparse matrix, common frameworks implement this as a multiplication of a sparse matrix and a dense matrix, as stated here: https://arxiv.org/pdf/1609.02907.pdf

---

### Decision · Program_Chairs · 2022-01-20

**Decision:**

Reject

**Comment:**

This paper deals with the important practical problem of speeding up GNNs.
Although the proposed method based on LSH may be considered to be a rather too simple preprocessing, it would be worthwhile to share the practical idea with the community as far as the proposed method is shown effective enough.
However, as pointed out by several reviewers, it is concerned that the experimental validation of this paper is not sufficient.
Further and deeper validations will make this paper stronger.